# Validation of an RF Image System for Real-Time Tracking Neurosurgical Tools

**DOI:** 10.3390/s22103845

**Published:** 2022-05-19

**Authors:** Carolina Blanco-Angulo, Andrea Martínez-Lozano, Carlos G. Juan, Roberto Gutiérrez-Mazón, Julia Arias-Rodríguez, Ernesto Ávila-Navarro, José M. Sabater-Navarro

**Affiliations:** School of Engineering of Elche, Miguel Hernández University of Elche, 03202 Elche, Spain; cblanco@umh.es (C.B.-A.); andrea.martinezl@umh.es (A.M.-L.); carlos.juan01@umh.es (C.G.J.); roberto.gutierrez@umh.es (R.G.-M.); julia.arias@umh.es (J.A.-R.); eavila@umh.es (E.Á.-N.)

**Keywords:** microwave-based medical image, real-time tracking, RF antenna system, surgical navigation

## Abstract

A radio frequency (RF)-based system for surgical navigation is presented. Surgical navigation technologies are widely used nowadays for aiding the surgical team with many interventions. However, the currently available options still pose considerable limitations, such as line-of-sight occlusion prevention or restricted materials and equipment allowance. In this work, we suggest a different approach based on a microwave broadband antenna system. We combine techniques from microwave medical imaging, which can overcome the current limitations in surgical navigation technologies, and we propose methods to develop RF-based systems for real-time tracking neurosurgical tools. The design of the RF system to perform the measurements is shown and discussed, and two methods (Multiply and Sum and Delay Multiply and Sum) for building the medical images are analyzed. From these measurements, a surgical tool’s position tracking system is developed and experimentally assessed in an emulated surgical scenario. The reported results are coherent with other approaches found in the literature, while overcoming their main practical limitations. The discussion of the results discloses some hints on the validity of the system, the optimal configurations depending on the requirements, and the possibilities for future enhancements.

## 1. Introduction

The development of new technological tools designed for aid during surgical navigation tasks has raised the interest of a considerable number of research groups during recent years. Waelkens et al. [1] compared the surgical navigation to the Global Positioning System (GPS)-based navigation. Indeed, GPS-based navigation consists of the GPS-based detection of the user’s current position and the subsequent identification of the most suitable path to the target destination, whereas surgical navigation requires the detection of the surgical tool’s current position and the subsequent identification of the optimal route to the surgical target (e.g., tumor). To do that, the system uses clinical images of the area of interest taken before (pre-operative) or during (intra-operative) the intervention and guides the surgeon’s movements according to the surgical tool’s detected positions. Analogically to the GPS system in the GPS-based navigation, the accurate detection of the surgical tool’s current position in an intra-operative basis is of paramount importance for the correct guiding and tracking during surgical navigation. The subsystem devoted to track the surgical tool’s position is called ‘tracker’. Several technological solutions have been proposed to implement such a system, the optical and the electromagnetic approach being the most common ones.

The optical trackers are widely used in current interventions. In these systems, a reference object (usually referred to as ‘fiducial marker’) is attached to the tip of the tool (or to any other point of interest), so that it can be spatially tracked by a pair of stereoscopic cameras placed at a convenient position (usually attached to the ceiling of the room). By triangulation calculations, the camera system can compute the spatial coordinates of the detected markers, associated to the tool’s coordinates. The markers may be either passive (e.g., near-infrared reflectors) or active (e.g., LEDs). In recent years, these systems have been combined with augmented reality (AR) tools to provide more immersive handling during surgical navigation and training of medical procedures [2], and even for marker-less surgical guiding approaches [3]. Despite the progresses, challenges such as misalignments between the physical and the virtual objects are still to be faced [4], as well as inaccuracies during the intra-operative AR-based navigation [5]. These systems can provide highly accurate surgical tool tracking [6,7], but they have the drawback of requiring direct line-of-sight contact with the markers [8]. If a body (someone from the staff) or an object (another tool, a piece of equipment, etc.) hinders this contact, the track is lost, in addition to other specific limitations associated to intra-operative imaging resolution [9] and misalignments between pre-operative and intra-operative images and tracking [10]. A couple of examples of these systems currently available in the market are the Polaris^®^ system from NDI [11] or the custom systems based on OptiTrack Motion Capture [12].

The electromagnetic trackers provide a solution for the direct line-of-sight requirement. In this case, the markers are made of ensembles of small sensor coils usually housed in a small case, again attached to the point to be tracked. The tracker is available to detect the spatial location of these markers due to the variations in the electromagnetic field caused by their interaction with the field. This detection can be made even when there is no direct line-of-sight between the tracker and the marker [13], thereby allowing free movements of the surgical team, and making them suitable for operations with minimal incision. However, these systems show two main drawbacks. Firstly, their effective action field is considerably reduced in comparison with the optical ones [14], and their application is limited to operations involving small areas, such as otorhinolaryngological ones. Secondly, due to the magnetic nature of the system, the measurements made by the tracker to detect the position of the markers can be altered by magnetic field distortion caused by metallic objects and electrically powered equipment in the surgical scenario [13,15], which poses a considerable limitation on the materials and equipment (including the tools themselves) that can be used for the intervention. As an example, an electromagnetic-based system currently available in the market is the Aurora^®^ system from NDI [16]. Hybrid optical and electromagnetic tracking systems have also been proposed [13], even including augmented reality tools as well [17], albeit always keeping the above-mentioned limitations.

Given the associated limitations to each method, new technological solutions to overcome them are still being pursued. Among the available options, microwave imaging rises as an interesting alternative that can provide for continuous and non-invasive intra-operative surgical tracking while overcoming the line-of-sight and magnetic interaction problems [18]. This technique is based on the changes in the dielectric constant and dielectric losses between the different biological tissues involved. These variations in the permittivity can be seen and identified by analyzing the changes in the response of microwave antennas [19]. With these alterations in the dielectric properties of the tissues, their boundaries can be identified and the biomedical image can be built [20]. These techniques find wide use in several fields, such as cancer detection and management [21,22]. Considering the usual contrast in dielectric properties, these techniques could also be applied to detect both surgical targets (such as tumors) and surgical tools [23]. In this work, we study the feasibility and validation of a microwave antenna-based imaging system for intra-operative surgical navigation.

## 2. Materials and Methods

The proposed RF-based surgical navigation system is made up of two main aspects, namely the system hardware and the signal processing methods. Both of them will be examined in detail in the next subsections, as well as the calibration of the whole system.

### 2.1. System Hardware

Figure 1 (left) shows the microwave imaging system. Briefly, it is made up of 16 wideband twin antennas located at equally-spaced points throughout a circumference with a diameter 34 cm—i.e., one antenna every 22.5°—around the area where the object under study is located, in this case a cranium 3-D model. This configuration leaves a space of 20 cm to place the model in the center of the circumference. Each antenna is held by a 3-D-printed holding piece, and the 16 antenna–holder pairs are screwed onto the top face of a wood board. All the antennas are connected by means of a high-frequency switching network to Port 1 (the port under use) of a Vector Network Analyzer (VNA) to perform all measurements (Rhode and Schwarz ZNLE6). An Arduino Due microcontroller is used to control the electronic switches so that only one antenna is active at the same time. Both the switches and the control system are located in the bottom face of the wood board, at a sufficiently far position under the antennas in order to avoid interferences in the measurement process. The whole system is controlled with a computer which executes Python script that automates the entire process. This script is in charge of the communication with both the VNA and the microcontroller. It therefore controls which antenna is connected to the network analyzer at each time, as well as the measurement files transfer between the VNA and the computer. It is also in charge of loading the measurement data into a MATLAB script that performs the signal processing and generates the images for the surgical navigation. The different parts of the system will be further described in the next paragraphs.

The wideband antennas constitute one of the key parts of the measurement system. The antennas are the front part of the microwave imaging system since they are responsible for signal transmission and reception, i.e., for data acquisition. In this case, they must have wideband behavior to be able to transmit the frequency information of the narrow time-domain pulses (which are similar to radar pulses), which are the basis of the imaging system. In this sense, the so-called Vivaldi antennas are one of the most widely used options in microwave imaging systems, for they offer a very large bandwidth and are slightly directive, concentrating the radiation in their aperture pattern [24]. For this work, we designed a modified version of the usual Vivaldi antennas so that they better fit in the proposed application without losing their capabilities. For this reason, detailed antenna design and characterization deserve further subsections, coming next. 

Continuing with the description of the system, the switch network, used to select the active antenna at each moment, is made up of commercial SP4T RF switches. These switches ensure that only one antenna is active at each moment, and they activate them all consecutively, one after another. The designed microwave imaging system uses a total of 16 antennas, which means that five switches are needed for a proper connection of the antennas to the VNA. In the selected configuration, one of the switches acts as the central one, which selects one of the remaining four switches, being each one of them connected to four antennas. In particular, we used the ZSWA4-63DR+ switch from Mini-Circuits. It is a high-speed, low-losses switch, based on CMOS technology, which is internally adapted to 50 Ω in the 1 MHz to 6 GHz frequency range, making it perfectly suitable for the frequency and switching speed requirements of our imaging system. The selection of the active output is made with three control voltages, which must follow the truth table provided by the manufacturer in the switch’s datasheet, always avoiding not supported combinations.

The control and DC power supply subsystem manages the power supply and control voltages for the switches. All the system is powered with the voltage supplied by the computer’s USB port. The control signals for the switches are provided by an AT91SAM3X8E microprocessor, embedded in the Arduino Due platform. DB9 connectors were used to connect the supply voltage and the control voltages of the switches with a four-core twisted shielded cable to reduce noise and interference. Control and power supply circuits were designed in order to make possible the connection between the microcontroller and the switch connectors, as well as a series of LED diodes were included to indicate the selected antenna at each moment. With this setup, the emitted power by the antennas is less than 1 mW. Finally, we used SMA connectors and the required coaxial cables to implement the different connections between the antennas, the switch RF-ports, and the VNA. Figure 1 (right) shows the final connection between the control circuit and the switches as well as the wiring system.

#### 2.1.1. Antenna Design and Standard Characterization in the Frequency Domain

The main drawback of Vivaldi antennas is their physical size, which is considerably large. This may pose a limit for its use in this type of imaging system, where a reduced size for the antennas is desirable. Vivaldi antennas are based on an exponential-profile radiating slot that features the same characteristics regardless of the frequency, so the bandwidth is theoretically infinite. In the design process, the size of the initial and final aperture is selected depending on the targeted frequency range of operation, being the size of the small aperture a half wavelength of the highest frequency and vice-versa. This aspect is especially important when determining the minimum operating frequency, that limits the size of the antenna. We, therefore, adapted the design to the above-mentioned frequency requirements. The feeding of the antenna is implemented with a microstrip transmission in the bottom face. For the impedance matching to 50 Ω throughout the entire frequency range, an open-ended radial stub transmission line is used with some transmission line sections with variable width [25]. Figure 2 shows a picture of the Vivaldi-like designed antennas, including the top and bottom layers.

The proposed antennas are a modification of classical Vivaldi antennas. To overcome the size restrictions, the final size was reduced by truncating the exponential profile of the slots and by modifying the geometric shape of the antenna aperture. This modification was made by optimizing the dimensions of the new aperture by simulations with ANSYS HFSS and Keysight EMPro software, so that the smallest antenna size is obtained while the frequency-domain performance of the antenna is unaltered. The main effect of this size-reduction process is the bandwidth decrease. In the design process, we pursued the best antenna characteristics between 0.1 and 6 GHz, according to the target frequency range. Additionally, three director elements were added in the antenna aperture region (see Figure 2) with the aim to increase the directivity of the antenna by concentrating the radiation pattern in the aperture’s direction. The final size of the antenna is 70 × 68 mm^2^, which means a size reduction of more than 4 times in comparison with a standard Vivaldi antenna with the same frequency characteristics. Finally, the antenna was printed onto a 1.52-mm-thick piece of FR4 substrate (with dielectric constant of 4.4 and loss tangent of 0.02) following a photolithography and chemical etching process.

One of the antennas was placed inside an anechoic chamber to carry out a standard characterization. Firstly, the return losses were characterized by means of the scattering parameter *S*_11_. The results of both the simulated and measured response are plotted together in Figure 3. An acceptable agreement can be seen, thereby validating the design, optimization, and implementation processes. The experimental operating frequency range of the antenna is from 1.2 to 5 GHz (for *S*_11_ < −10 dB), which means 123% bandwidth for a central frequency of 3.1 GHz; thus, confirming its wideband nature. Secondly, the antenna radiation patterns were measured by employing a horn antenna as a reference. The measurement characterization of the antenna in terms of the E-plane and H-plane radiation diagrams at specific frequency points can be seen in Figure 4. It is worth noting that the designed antenna has a higher directivity in the E-plane (the aperture plane), especially at higher frequencies. Following these steps, the 16 involved antennas were characterized, obtaining well-nigh identical results.

#### 2.1.2. Antenna Time Domain Analysis

The signals used in medical imaging systems are usually broadband, i.e., they involve remarkably short pulses in time covering a considerably broad frequency spectrum. Since the pulses are narrow, they are greatly affected by dispersion. As a result, the incoming pulse at the antenna will never be the same as the outgoing pulse. For this reason, a time domain analysis of the transmitted pulses was completed, in order to predict the distortion inherent to the system.

To do that, the transfer function for a system in which a pulse is transmitted and received by the same single antenna (like the system shown here) can be measured and characterized by an equivalent system in which there is a direct transmission between two identical antennas (i.e., the receiving antenna is exactly the same as the transmitting antenna). This assumption considerably simplifies the analysis and measurement of the transfer function of the original system, and it will be therefore taken in this section. Please note that this assumption is taken for characterization purposes only. This scheme uses two identical antennas vertically oriented in a face-to-face manner and separated by 40 cm, distance sufficient to ensure far field transmission. This way, the system’s transfer function reduces to the *S*_21_ parameter. Therefore, the two antennas were located inside an anechoic chamber and the *S*_21_ parameter was measured using the VNA, with one antenna acting as a transmitter while the other as a receiver. The experiment was replicated in silico by means of free-space transmission simulations. The information obtained after this frequency-domain analysis was post-processed to obtain the time domain signals.

From the *S*_21_, some metrics can be used to analyze the performance of the antenna, such as the group delay (i.e., the phase derivative for the *S_21_* response). The experimentally measured and simulated group delays for the above-mentioned antenna arrangement are plotted in Figure 5. It can be seen that it is fairly flat within the antenna bandwidth (shadowed area), with values between 0.20 and 0.25 ns. These results suggest that the system will show a low distortion for signals whose bandwidth falls within the antenna bandwidth. The best way to quantify this effect is to use the so-called System Fidelity Factor (SFF), which is a measurement of the correlation between the transmitted and received pulses. This factor calculates the ratio between the energy of the convolution between the transmitted and received pulses and the energy of each pulse separately [26]. The SFF is defined as:(1)$$\mathrm{SFF}=max_{n}\left|\frac{\int_{-\infty }^{+\infty }T_{S}\left(t\right)R_{S}\left(t+\tau \right)d\tau }{\sqrt{\int_{-\infty }^{+\infty }{\left|T_{S}\left(t\right)\right|}^{2}dt}\cdot \sqrt{\int_{-\infty }^{+\infty }{\left|R_{S}\left(t\right)\right|}^{2}dt}}\right|$$
where *t* is the time, *T_S_* is the transmitted pulse, and *R_S_* is the received pulse, which is computed from the standard *S*_21_ parameter (which means that SFF takes into consideration the distortion induced by both antennas). Thus, for the proposed antennas the SFF value obtained is 95.29% if the theoretical transfer function is used, and 96.97% if the measured *S*_21_ is used, which gives an idea of the high signal integrity achieved for the transmitted signals.

### 2.2. Signal Processing for Imaging

After having described the hardware components of the system, we will focus on the algorithms used to obtain the images for the surgical navigation system from the measured information. A Python script that manages all the communications with the VNA and the Arduino microcontroller was created for the measurement process. On the one hand, the communications with the VNA are made according to the TCP-IP protocol through a laboratory local area network (LAN). To that end, a set of instructions were defined using the VISA (Virtual Instrument Software Architecture) protocol. On the other hand, the communication with the microcontroller uses a serial communication channel via USB.

To eliminate, as much as possible, the interferences and reflections caused by the switches, cables, and other electronic elements of the system, as well as the offset of the equipment, an individual SOL (short–open–load) calibration for each antenna is carried out with the VNA before any measurement. The calibration is made once all the antennas and equipment are placed in their corresponding places and properly connected. This calibration is saved and later loaded for each antenna during the measurement process, taking into account the active antenna at each moment. In addition, to reduce the effect of the different elements composing the hardware system, such as the rest of the antennas or the switches (as well as further electronic equipment), a reference measurement for each of the antennas is carried out. This measurement will be used during the signal processing to eliminate reflections that are not strictly due to the bodies to be tracked with the aim to increase the sensitivity and accuracy of the system.

In this work we study the use of two different algorithms for building the medical image: Delay and Sum (DAS) and Delay Multiply and Sum (DMAS). The following subsections will provide an overview of each of them.

#### 2.2.1. Delay and Sum

The direct analysis of the raw measured responses of the antennas can be useful for the fast detection of simple objects, but it is not enough for building complex and detailed images of the scenario under tracking. However, this is crucial in the case of assistance to surgical intervention, where the produced images must be interpreted from the spatial point of view to indicate where the tool and the interesting objects are located. In these cases, the use of further data processing algorithms that allow a clearer and more concise representation of the information and that allow the representation of the objects and materials found inside the scenario is essential. One of these algorithms is known as Delay and Sum [27]. This algorithm consists of carrying out a spatial modeling of the elements to be analyzed. In this model, the position of each of the antennas is known, and the model accordingly divided into a grid. Each of the vertices of the grid is a calculation point where the signal delay from each of the antennas to that specific point is computed and later a weighted sum of the results is calculated, yielding the resulting intensity for each point of the image. The analytical formulation for DAS algorithm is the following:(2)$$I\left(r_{0}\right)=\int_{0}^{T_{Win}}{\left[\overset{M}{\underset{m=1}{\sum }}X_{m}\left(r_{m}\left(r_{0}\right)\right)\right]}^{2}dt$$
where *I*(*r*_0_) is the obtained intensity for the point *r*_0_ of the grid, *M* is the total number of antennas, *T_Win_* is the integration window, which is usually considered as a percentage of the input pulse signal, and *X_m_* is the time-domain signal obtained with each antenna, which has a certain delay in the point *r*_0_. The delay in each point, *r_m_*_,_ is obtained with the following expression:(3)$$r_{m}\left(r_{0}\right)=\frac{d_{m}}{v}f_{s}$$
being *d_m_* the distance between the antenna and the point *r*_0_, *f_s_* the sample frequency of the signal (which is used to obtain the Inverse Fourier Transform for the analysis), and *v* the propagation speed of the medium, which is given by:(4)$$v=\frac{c}{\sqrt{\varepsilon_{r}}}$$
where *c* is the speed of light in a vacuum and *ε_r_* is the dielectric constant of the materials through which the wave travels. This means that the DAS algorithm needs to know the dielectric properties of the medium, or at least an estimation of them, to accurately determine the signal delays of each antenna. The precision of the algorithm therefore depends on how good the dielectric constant estimation is.

#### 2.2.2. Delay Multiply and Sum

DMAS algorithm [28] is a variant of the DAS algorithm in which a multiplication of the calculated delays is performed for each pair of signals obtained using the measurement system. This way, the coherence or correlation between the reflections and the precision in the representation of the images are increased. Being ⊙ the element-wise product operator, the analytical formulation for DMAS is:(5)$$I\left(r_{0}\right)=\int_{0}^{T_{Win}}{\left[\overset{M-1}{\underset{m=1}{\sum }}\overset{M}{\underset{j=\left(m+1\right)}{\sum }}X_{m}\left(r_{m}\left(r_{0}\right)\right)⊙X_{j}\left(r_{j}\left(r_{0}\right)\right)\right]}^{2}dt$$

### 2.3. Calibration of the Imaging System

Once the hardware system and the algorithms used to obtain the image of the elements under study have been described, the tests carried out for the start-up and calibration of the system will be presented in this subsection. The calibration of the imaging algorithms was based on measurements and imaging of a metallic cylinder of 5.0 cm diameter and 11.5 cm height. Figure 6 shows a picture of the cylinder together with the antenna system.

After placing the metallic cylinder in the center of the structure, equidistant from all the antennas, the measurements and the processing of the obtained data were carried out. The resulting images for DAS and DMAS algorithms are shown in Figure 7. As it can be seen, an image of the metallic cylinder edge is obtained with a high intensity representation, in yellow color, corresponding to its real dimensions. It can also be seen how some light-blue-colored areas appear, which are related to the signal reflected by the cylinder and rebounded in the metallic parts of the neighbor antennas before being received by the active antenna at each moment. These reflections are more attenuated in the DMAS algorithm, which offers a clearer image (and shape) of the object for the considered setup.

## 3. Experimental Validation and Results

### 3.1. Experimental Setup

Once the system was implemented and calibrated, its performance as a cranial surgery navigation tool was assessed. The targeted application consists of detecting the position of the surgical tool within the cranial area, so that the surgical team is provided with proper guidance and assistance. In order to emulate such a scenario, a setup imitating an operating room for cranial surgical interventions with robotic tools was prepared, as shown in Figure 8. As it can be seen, a structure was designed to hold the antenna system, allowing to house the 3-D-printed cranium in the exact center. Ad hoc connection of the antennas to the VNA and the computer was installed, as explained in the prior sections, to allow the proper running of the microwave image system. Finally, a UR5 robotic arm was used to emulate the surgeon’s moves. The robotic arm was configured to hold the clinical tool, which is intended to navigate towards the critical area within the cranium.

For the experimental validation, an accurate and detailed 3-D-printed model for the cranium was used (horizontal section dimensions 128 × 170 mm^2^), which had a hole on the left side of the forehead to enable the entrance of the surgical tool, thereby allowing us to simulate intracranial surgery. A piece of plastic filled with water was used to imitate a tumor, which was placed inside the cranium, in the inner center forehead area (Figure 9). The cranium was placed in the center of the antenna system, so that the cranium’s center coincided with the center of the antenna coordinate system, as can be seen in the picture. With this configuration, the tumor was slightly displaced from the center of the coordinate system, with the tumor’s center position at roughly (0, 26) mm coordinates considering the framework in Figure 9. The tumor had 25 × 25 × 25 mm^3^ dimensions.

The navigation sequence for the experimental validation consisted of the surgical tool being approached to the tumor, throughout the cranium’s hole, following a spatially diagonal straight line (also including the approach in the *z* coordinate, which is not considered in the 2-D images provided by the microwave image system) at the same time that the microwave image system was tracking the tool position. The tool path started at a position sufficiently far from the tumor (more than 150 mm away), and it was planned to finish as close as possible to the tumor’s center; thus resembling the trajectory followed during an actual intervention. Although the tool may be handled by the surgeon in real operations, the robotic arm was used to hold it during this experiment so that the trajectory and navigation sequence could be accurately controlled, and reference coordinates throughout the trajectory could be obtained in a reliable manner. A picture of the tool entering the cranium and the antenna system can be seen in Figure 10.

### 3.2. Image Acquisition

Before starting the experiment, a measurement only with the antennas, with nothing inside the system (empty measurement) was made. This empty measurement was used as the reference and calibration measurement throughout the whole experiment, and the rest of images presented in this work included the subtraction of this empty measurement. Then, two initial measurements were made with the microwave image system for the proposed setup: one only having the cranium (no tumor, no tool), and another one having the cranium and the tumor (no tool) in the right positions. During the navigation sequence, 8 measurements were made with the microwave image system at 8 different moments, so that the 8 corresponding images could be assessed. Parallell to each of them, the coordinates of the robotic arm at each moment were saved, which were later transformed to tool’s final-end coordinates for reference. Within these 8 positions, hereinafter referred to as p*i*, where 1 ≤ *i* ≤ 8, p1 and p2 had the tool’s final end out of the antenna system space, p3 and p4 had it inside the antenna system space but out of the cranium, p5 had it approximately in the cranium boundary (entering the hole), p6 and p7 had it inside the cranium and gradually approaching the tumor, and p8 had it inside the cranium and well-nigh touching the tumor boundary. Finally, the 10 resulting measurements (2 initial ones + 8 during the tool’s navigation) were processed with both DAS and DMAS algorithms. The flowchart for the imaging algorithm applied to each measurement is depicted in Figure 11.

The resulting images for both DAS and DMAS algorithms for the 2 initial measurements are shown in Figure 12, whereas the corresponding images for a selection of some of the 8 navigation measurements can be seen in Figure 13 (positions 1 and 2 are not shown since the tool was out of the antenna system area). The images in Figure 13 show the evolution of the tool’s final-end position during the navigation experiment, although some other reflections are detected due to the long-shaped tool’s body. Considering a long enough tool (which is the most usual case in these operations), these reflections are approximately constant for two consecutive images or positions (provided that a low or moderate differential movement was made), the new information only being related to the position change. Therefore, aiming at a better detection of the tool’s final-end position for navigation purposes, the images were further processed by subtracting the previous image from the current one, so that only the tool’s displacement information was left, and the new position can be easily tracked. The resulting images for both algorithms are shown in Figure 14, again excluding positions 1 and 2 for the same reason.

### 3.3. Data Extraction and Results

The images in Figure 14 show a clear evolution of the tool’s final-end position throughout the trajectory. These images are, therefore, suitable for providing a targeted navigation track. To provide navigation assistance and accurate guidance capabilities, the exact coordinates of the tool’s position must be detected. To that end, each image in Figure 14 was binarized with a 0.8 threshold, which means that a new binary black-and-white associated image was created in which the pixels with luminance lower than 0.8 in the original image were set to black, otherwise to white. An example of this process for “p5–p4” images for both algorithms is shown in Figure 15. As it can be seen, the resulting images are more convenient for processing and detection of properties. Each binarized image was then analyzed and the coordinates of the centroid of the remaining white region were computed. As seen in Figure 14, the high-luminance regions correspond to the tool’s final-end positions, and therefore these centroids’ coordinates were associated to the real tool’s final-end coordinates.

The binarized images had 1300 × 1301 dimensions. The computed centroids were defined within an image-based coordinate system, having the origin in the top-left corner. In order to be able to compare with the reference coordinates of the tool’s final end, which were obtained through the robotic arm positions, considering the common coordinate system centered in the antenna system (and in the cranium as well), the equivalence of pixels to physical distance was required. These data were obtained thanks to the real cranium dimensions and its pixel-based dimensions computed from the initial cranium images (see Figure 12 top). The equivalence was thereby found to be 1 column pixel = 0.1922 mm; 1 row pixel = 0.2255 mm. With these data, the microwave image-based detected tool’s final-end coordinates were obtained. A scheme of the coordinate detection process is depicted in Figure 16.

The results for the detection of the tool’s final-end coordinates are shown in Table 1, which gathers the detected coordinates both with DAS and DMAS algorithms when the differential images (Figure 14) are considered and compares them to the reference coordinates obtained from the robotic arm at each position. Position p1 is not considered since there was no prior position to perform the image subtraction and the tool was considerably far from the antenna system. The evolution of the detected coordinates with both algorithms, as well as the reference coordinates from the robot, show the approaching trajectory of the tool to the tumor, from p2 to p8. Specifically, the detected coordinates in p8 for DAS and DMAS algorithms show a difference of (12.5886, −15.1963) mm and (12.3963, −15.7374) mm, respectively, with respect to the corresponding detected coordinates of the tumor’s center (first row), which are coherent with the visual observation (see Figure 17). This is also confirmed by the small error in p8 for DAS and DMAS with respect to the reference coordinates from the robot: ∆(p8) = (−0.8544, −0.5576) mm for DAS and ∆(p8) = (−1.3925, −0.1743) mm for DMAS.

Considering the reference coordinates from the robot, Table 2 shows an error analysis for the performance of DAS and DMAS. Position p2 was excluded because the tool was still too far from the antenna system region. The mean error ($\bar{\Delta }$) and the standard deviation (*σ*) for each coordinate (*x* and *y*) were independently considered in this analysis. Additionally, different ranges of positions are involved, depending on the different regions where the tool navigated: p3 to p8 includes the tool travelling through the antenna system inner space, the cranium boundary, and the cranium inner space, p5 to p8 includes the tool travelling through the cranium boundary and its inner space, and p6 to p8 includes only the tool travelling through the cranium inner space.

Finally, the influence of the luminance threshold was analyzed. Given the agreement between the detected coordinates with both algorithms at p8 and the reference coordinates from the robot arm, this position was taken as general reference. The detected coordinates were recalculated from the saved measurements with DAS and DMAS algorithms using different luminance thresholds. For each newly recalculated pair of coordinates (associated to a certain luminance threshold), the detection error (*DE*) was computed as the Euclidean distance to the reference coordinates:(6)$$DE=\sqrt{{\left|x_{d}-x_{r}\right|}^{2}+{\left|y_{d}-y_{r}\right|}^{2}}$$
where *x_d_* and *y_d_* are the detected coordinates, whilst *x_r_* and *y_r_* are the reference coordinates. The resulting evolution of the *DE* depending on the luminance threshold for DAS and DMAS algorithms at p8 is plotted in Figure 18. It should be noted that this plot was made from only one measurement (p8), which was analyzed many times with different luminance thresholds. It therefore gives information about the digital error when computing the coordinates as a function of this threshold.

## 4. Discussion

A microwave-based image system for cranial intraoperative tool navigation was proposed, and its performance was assessed. The system is composed of 16 twin Vivaldi-like antennas placed throughout a circumference with an equally spaced pattern, surrounding the cranial surgery area, pointing to the center of the circumference. An automated switching electronic system is used to drive the antennas and make the corresponding reflection measurements. The responses of the antennas are affected by the reflections of the electromagnetic waves on the cranium shape and on strange objects, such as tumors or surgical tools. These responses are further processed to locate the desired objects and provide surgical tool navigation. It should be noted that the maximum emitted power by the antennas is lower than 1 mW, which is less than the usual power involved in a cellphone call [29]. The proposed system is thereby suitable for use in clinical scenarios.

Two methods were studied to process the responses of the antennas and build the medical image. These methods, viz. DAS and DMAS, consist of a spatial modeling of the surgical environment by assigning a computed intensity to each pixel of the image depending on the corresponding formulation and the time-domain response of each antenna. The following paragraphs will discuss the experimental validation and results of the proposed system using these two methods.

Figure 12 shows the capability of the proposed system to scan the cranium and detect the tumor within the experimental setup considered here. Both algorithms show acceptable detection capabilities in this regard. Considering the images in which only the cranium is involved (Figure 12 top), the DAS algorithm provides brighter images, which allow us to see a higher level of detail. This should be analyzed with caution, because it also implies the apparition of spurious details, such as the reflected beams captured by each antenna, which do not correspond to any physical object in the scenario. That being said, as long as the spurious information is static and previously known (such as these beams, directly related to the position of each antenna), it could be easily eliminated. The DMAS algorithm, however, provides a cleaner image, almost with no spurious details, but with a more poorly defined cranium. Conversely, when the tumor is involved (Figure 12 bottom), DMAS seems to show better detection capabilities, providing a clearer, more defined location of the tumor. In this case, the high-intensity reflections by the tumor material (in comparison to those by the cranium material) hinder the detection of the cranium shape in both methods, being less visible (but detectable) for the DAS algorithm and almost invisible for the DMAS algorithm. Considering these pictures, it seems that both algorithms show strengths and weaknesses for different aspects, and therefore a detailed analysis for both of them is worthwhile. Ostensibly, the DAS image can be more suitable for calibration tasks, for example, taking reference measurements of the cranium’s dimensions, and also for detection and tracking of events within the cranium area, which is better resolved with this algorithm, whereas DMAS seems to show better performance regarding accurate location of strange objects within the image framework, although losing information related to the cranium shape.

In the experimental results for the navigation task shown in Figure 13, it can be seen how the long shape of the tool hinders the clear and direct identification of the tool’s final-end position, especially for the last positions (the tool entering the cranium, longer tool’s body portion within the image). Indeed, the long-shaped tool yields the detection of many reflections throughout the tool’s body by different antennas, depending on their position. This information could be useful for further processing of the images in the final system, so that the full shape of the tool can be depicted in the image shown in the user-oriented graphic interface. However, for the pursued navigation assistance, considering the binarization process proposed here, this phenomenon leads to the apparition in the binarized images of several areas with several associated centroids, and the detection of the current position of the tool becomes complicated. In addition, other objects different from the tool could be detected, leading to the definition of false positions for the tool. For example, it can be seen that the tumor is detected in positions p1 and p2 with both methods, since the tool had not yet arrived at the detection area at those moments. Therefore, for navigations purposes, we propose the differential method in which the prior image is subtracted to the current one, as shown in Figure 14, so that the undesired, unmoved details are eliminated and only the information related to the tool’s trajectory evolution is tracked.

Figure 14 shows the images obtained with the differential method for navigation purposes, for both algorithms. Here, the information obtained from each image is only related to the tool’s trajectory, i.e., the difference in the tool’s final-end position between the last measurement and the current one. These images provide a clear view of the trajectory followed by the tool, starting out of the measurement area and following a straight line towards approximately the tumor’s position. This approximation can be seen by observation of the images “p8–p7” in Figure 14 and the bottom images in Figure 12. The proposed process, including the binarization of the resulting image and the computation of the centroid in the high-luminance region, allows the detection of the tool’s final-end coordinates in the current position, thereby tracking the tool’s navigation. The results for this position detection process again confirm the approach of the tool to the tumor’s location, as can be seen in Table 1. Considering these results, it should be noted that: (1) the tumor’s position coordinates refer to the tumor’s exact center, which cannot be physically reached by the tool in the proposed setup due to the physical dimensions of the solid object emulating the tumor; and (2) the tool in position p2 was out of the measurement range, and no information can be obtained from this position.

The comparison between the detected positions with both algorithms and the reference positions obtained from the robot’s coordinates (Table 1 and Table 2) shows a good agreement, and it therefore confirms the potential of the proposed system for intraoperative navigation imaging. The detected positions and the error analysis yield similar results for both algorithms. The error analysis results show smaller errors and standard deviations for both algorithms for the innermost region. This is coherent with the detected positions, in which the closer the tool is to the tumor’s position, the smaller the difference between the detected position and the reference one. With the tumor (and the innermost region) being close to the center of the coordinates, this means that the error becomes smaller as the detected positions approach the center, which is logical given the radial configuration for the antenna system. As a consequence, the highest accuracy will be achieved for the innermost positions of the tool, located within the cranial area, meaning that the system is optimized for higher accuracy and resolution in the most interesting region for cranial surgery.

In this regard, the system shows a mean error of roughly 1.26 mm in the best case and 3.02 mm in the worst case for the interesting region with respect to the reference coordinates. Considering the cranium total dimensions, this means errors between 0.98% (best case) and 1.78% (worst case). It should be noted that, for magnetic-based tracking systems, mean detection errors of ~0.5 ± 0.5 mm have been reported [30], which can raise up to 27 mm due to interference of metallic objects [8]. For optical tracking, mean errors of 0.24 ± 1.05 mm have been reported, which can raise up to 1.65 ± 5.07 mm when some cameras are occluded [7]. The detection errors reported here are also consistent with the errors reported in other microwave imaging approaches. For example, a similar system was used in [31], also with 16 antennas (operating at 1–4 GHz), to detect intraoperative cranial inner hemorrhages, which reported detection errors between 1 and 5 mm. With the positioning error being dependent on the wavelength of the highest frequency in the system (which is linked to the resolution), the reported results here show consistency with those in [31].

Figure 14 also shows that, after the previous image subtraction, the DAS algorithm does not provide a graphical view of the tool’s final-end position in a manner as clear and well-defined as the DMAS algorithm does. Notwithstanding that, in this case, given the simple shape of the tool, the results regarding the position detection after the binarization and centroid computation process are quite similar for both algorithms, as shown in Table 1. That being said, the visual inspection of Figure 14 suggests that the DAS algorithm is more sensible for the luminance threshold (kept constant at 0.8 throughout the whole results analysis). Indeed, lower thresholds would have resulted in a sort of half-moon-shaped white area in the binarized images, instead of the ellipsoid-shaped ones for 0.8, as shown in Figure 15. With the centroids being computed as the mass center of the white area, a lower threshold would lead to a displacement of the finally detected position; thus yielding to a greater error in the detection. Figure 14 confirms that this phenomenon is considerably less noticeable for the DMAS algorithm. The analysis of the detection error as a function of the luminance threshold is shown in Figure 18, which confirms this behavior. These results highlight the dependence of DAS on the luminance threshold and allow us to conclude that DMAS is more robust to DAS to variations in this parameter. Consequently, DMAS is expected to show a more reliable performance when tools with more complex shapes are considered, or when rotations of the tool are involved.

Apart from this criterion, no further reasons were detected to claim the outperformance of one algorithm with respect to the other one. It should be noted that the setup considered in this study inherently has a certain instrumental error regarding the reference coordinates obtained from the position of the robot, due to the vibrations of the links of the robot during the movement as well as the oscillations of the tool’s final end due to its long shape and the tip-based holding. Therefore, seeing the small differences in the performance of both algorithms (see Table 1 and Table 2), both algorithms show acceptable performance for intraoperative tool navigation tracking, and we cannot point to any algorithm being the most advantageous regarding the detection accuracy for the experimental setup considered in this study and a properly selected luminance threshold. 

As a matter of fact, the raw images (before applying the differential method) for both algorithms (Figure 13) show similar information and even similar shapes for the high-luminance areas, and therefore this above-mentioned higher robustness of DMAS to the luminance threshold seems to come from the differential stage. It should be noted that the DMAS formulation inherently implies noise filtering, which often means detail loss. DMAS differential images (Figure 14) can provide a robust tracking of the tool’s final end, but the information related to the tumor position is blurred. DAS raw images (Figure 13), however, allow to see the tumor and even the cranium shape in addition to the tool, which would allow for intraoperative tracking of the tumor. This is a highly desirable feature, and therefore the combination of the information extracted from both algorithms could provide the surgical team with highly accurate intraoperative navigation and guidance for the approach of the surgical tools to the tumor position, even when changes in the tumor’s position are involved, such as those resulting from the brain-shift effect. It should be noted that this would be made only at the expense of a slightly higher computational cost, with no extra hardware required, since both algorithms would independently process the same measurements. The reported system, combining both algorithms, is thereby proposed as a potential surgical navigation system to robustly address interventions prone to tumor displacements.

The discussion shown in this initial study should be, however, limited by the restricted validity of the experimental setup, notably simplified. For a more realistic scenario, including real biological tissues or phantoms mimicking them, the dielectric properties of the materials involved would be different, and the measurements would be thereby altered. The measurements made with each antenna, upon which the images are built, are based on the reflections of the emitted electromagnetic waves while travelling through the scenario. These reflections occur when the waves travel through the boundaries of consecutive mediums with different dielectric properties. They mostly depend on the dielectric constant and conductivity differences in the boundary, rather than on the specific values for each medium. Although the dielectric properties for real biological tissues are evidently not the same as in the proposed experimental setup, the performance of the proposed system can be predicted by their differences.

Focusing on the tumor detection and tracking tasks, the average dielectric constant for health tissues in the brain is approximately 42, whereas for tumor tissues it turns to roughly 55 due to the high water content [32,33]. This means a relative increase of 30%, which is a sufficient difference to allow the tumor detection by means of microwave imaging techniques. It should be noticed that these techniques have been reported to handle and detect accurately contrasts as low as 4% [22]. As for the surgical tool, the evident differences in the materials (chiefly metals vs. biological tissues) and their properties allow to foresee good detection with the proposed system. All these differences allow to expect good detection capabilities both for the tumor and the tools when more realistic phantoms are involved, or even in real surgery scenarios, thereby potentially providing for the pursued RF-based real-time surgical tool tracking.

In addition, there are several strategies that could be applied in order to mitigate a hypothetic misperformance in a more realistic scenario, if required. More specifically, the properties of the tissues in a real-case brain, which allow the detection of tumors through the differences seen in the propagation speed of the electromagnetic waves travelling through them, are notably affected by their dielectric constant (*ε_r_*), as seen in (4). In this sense, some strategies could be applied for a more accurate detection. For example, the *ε_r_* of the materials could be characterized or estimated by means of initial measurements considering the *S_21_* parameter of active face-to-face antenna pairs. Additionally, already-known average values for the dielectric properties of biological tissues could be assumed. Another approach could be the use of filters and further processing techniques for the measured signals, so that more accurate detection of the properties of the materials the waves travel through could be achieved, and therefore a suitable propagation speed could be assigned for each case. Such filters could include, but are not limited to, adaptive beamforming algorithms [34] and hybrid methods [35]. It should be noted that these strategies are independent one to another, and there is no constraint that could prevent simultaneous use. Consequently, all of them could be used and combined in a proper way, so that the accuracy and detection capabilities could be enhanced as much as possible, attaining solutions adapted to each specific case.

## 5. Conclusions

An RF-based medical image system was proposed for surgical navigation tracking. This system, based on the reflections on electromagnetic waves emitted by the antennas, which are due to differences in the permittivity of the materials traveling through, arises as a potential option to overcome the usual limitations of current optical- or magnetic-based surgical navigation systems. The experimental assessment of the proposed system showed accuracies and errors consistent with other approaches with other technologies found in the literature; thus, highlighting the interest for further studies. The emitted power makes the system suitable for clinical use. The research on two imaging algorithms, DAS and DMAS, showed not enough evidence for claiming the outperformance of one over another. Indeed, we discussed in the interest of both of them, depending on the case, the environment characteristics and the target objects and tools. The combined use is, therefore, advised. As possible future work, we propose further experiments with more realistic, sophisticated biocompatible models for the cranium, the involved tissues, and the usual surgical targets. 

## Figures and Tables

**Figure 1 sensors-22-03845-f001:**
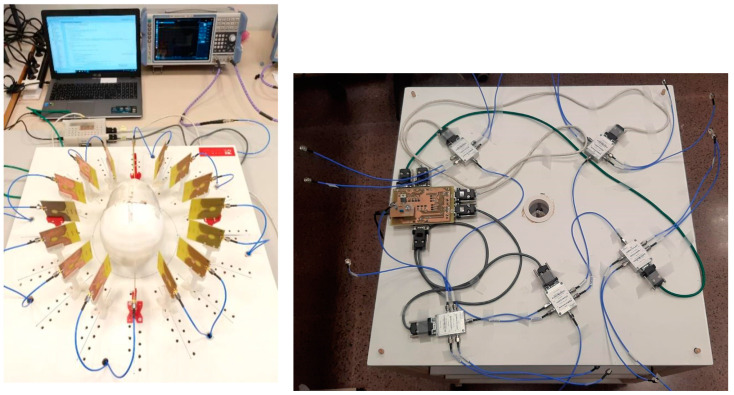
Microwave image system (**left**) and control and switching subsystem (**right**).

**Figure 2 sensors-22-03845-f002:**
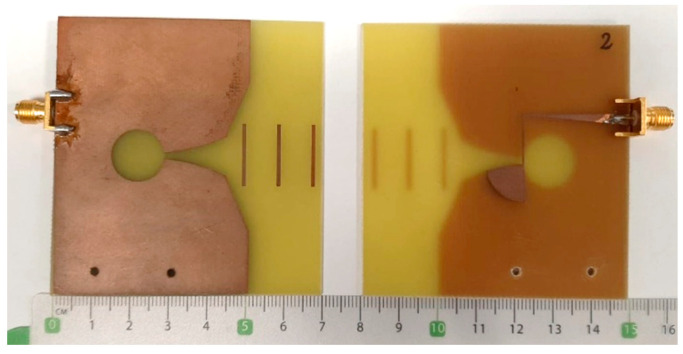
Designed Vivaldi-like antennas: top (**left**) and bottom (**right**).

**Figure 3 sensors-22-03845-f003:**
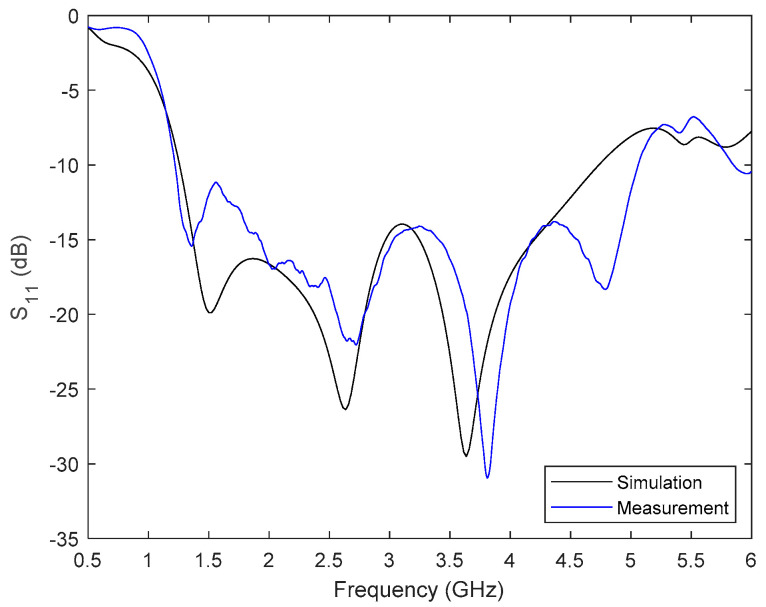
Measured and simulated return losses for the proposed antenna.

**Figure 4 sensors-22-03845-f004:**
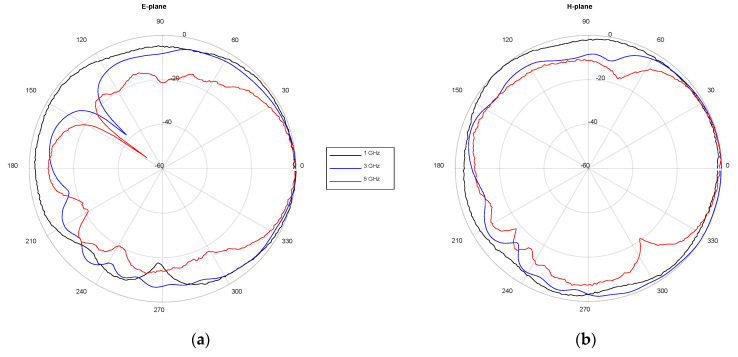
Measured antenna radiation pattern at specific frequency points: (**a**) E-plane; (**b**) H-plane.

**Figure 5 sensors-22-03845-f005:**
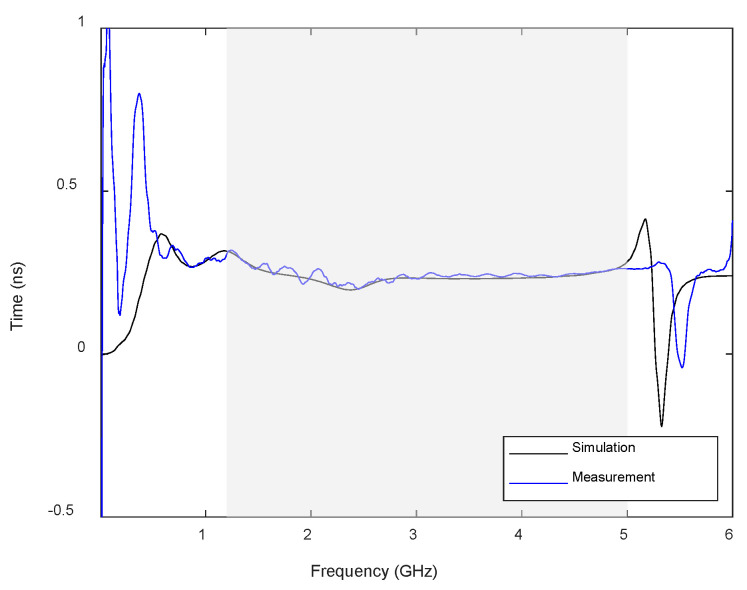
Measured and simulated group delay for the proposed antenna (the shadowed area indicates the antenna bandwidth).

**Figure 6 sensors-22-03845-f006:**
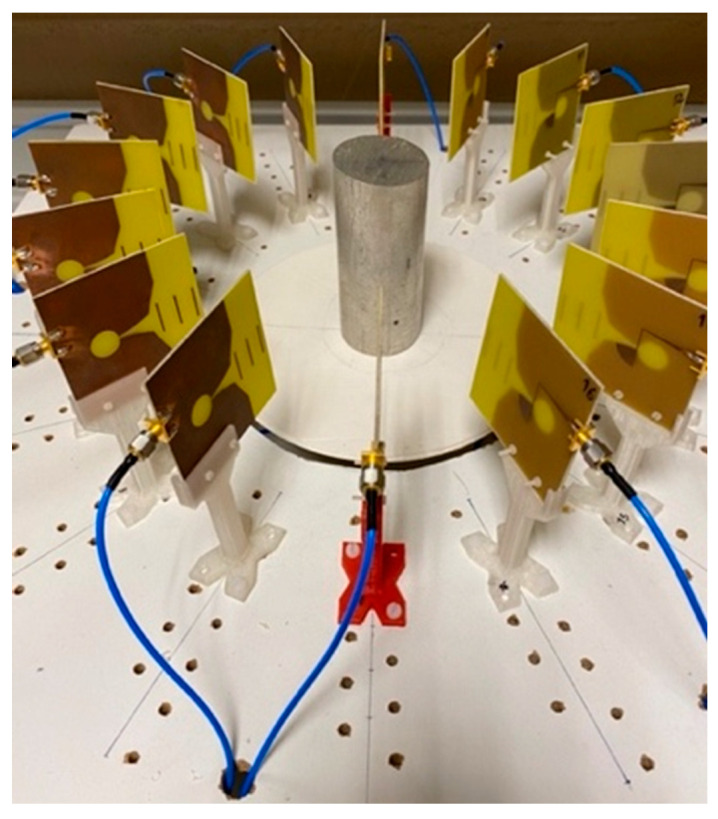
Calibration measurement with a metallic cylinder.

**Figure 7 sensors-22-03845-f007:**
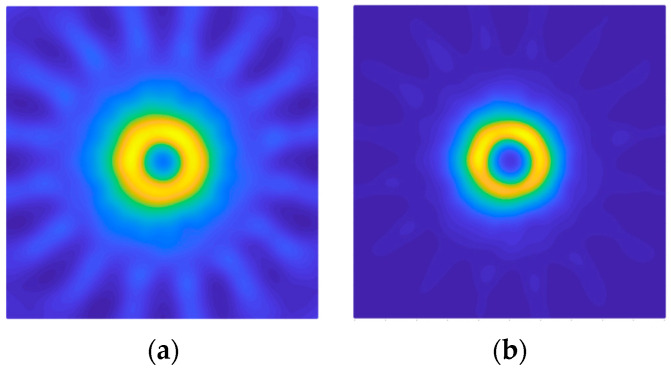
Obtained images for the measurement with the metallic cylinder: (**a**) DAS; (**b**) DMAS.

**Figure 8 sensors-22-03845-f008:**
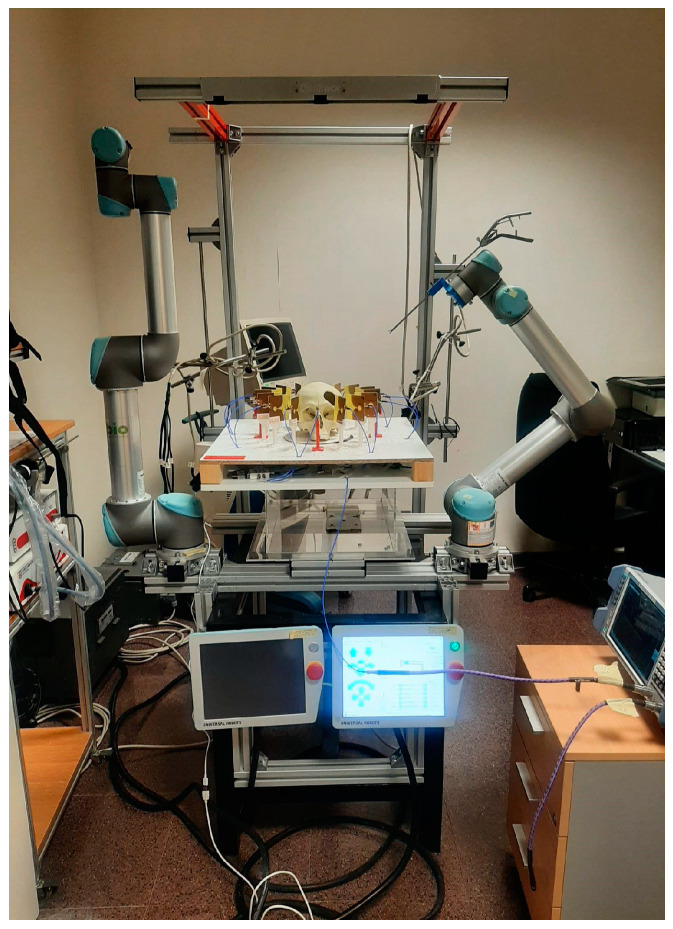
Experimental setup.

**Figure 9 sensors-22-03845-f009:**
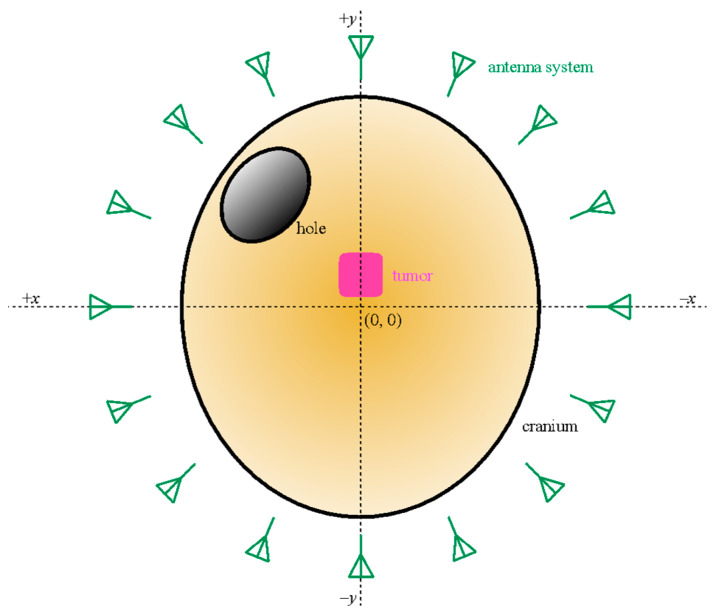
Schematic representation of the cranium (yellow), the hole (black), the tumor (pink), the antennas (green), and the coordinate system.

**Figure 10 sensors-22-03845-f010:**
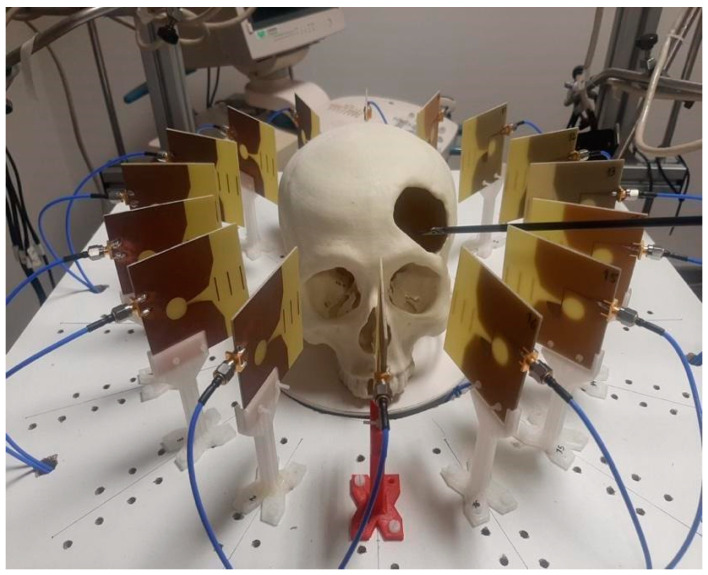
Experimental setup with the tool entering the cranium within the antenna system.

**Figure 11 sensors-22-03845-f011:**
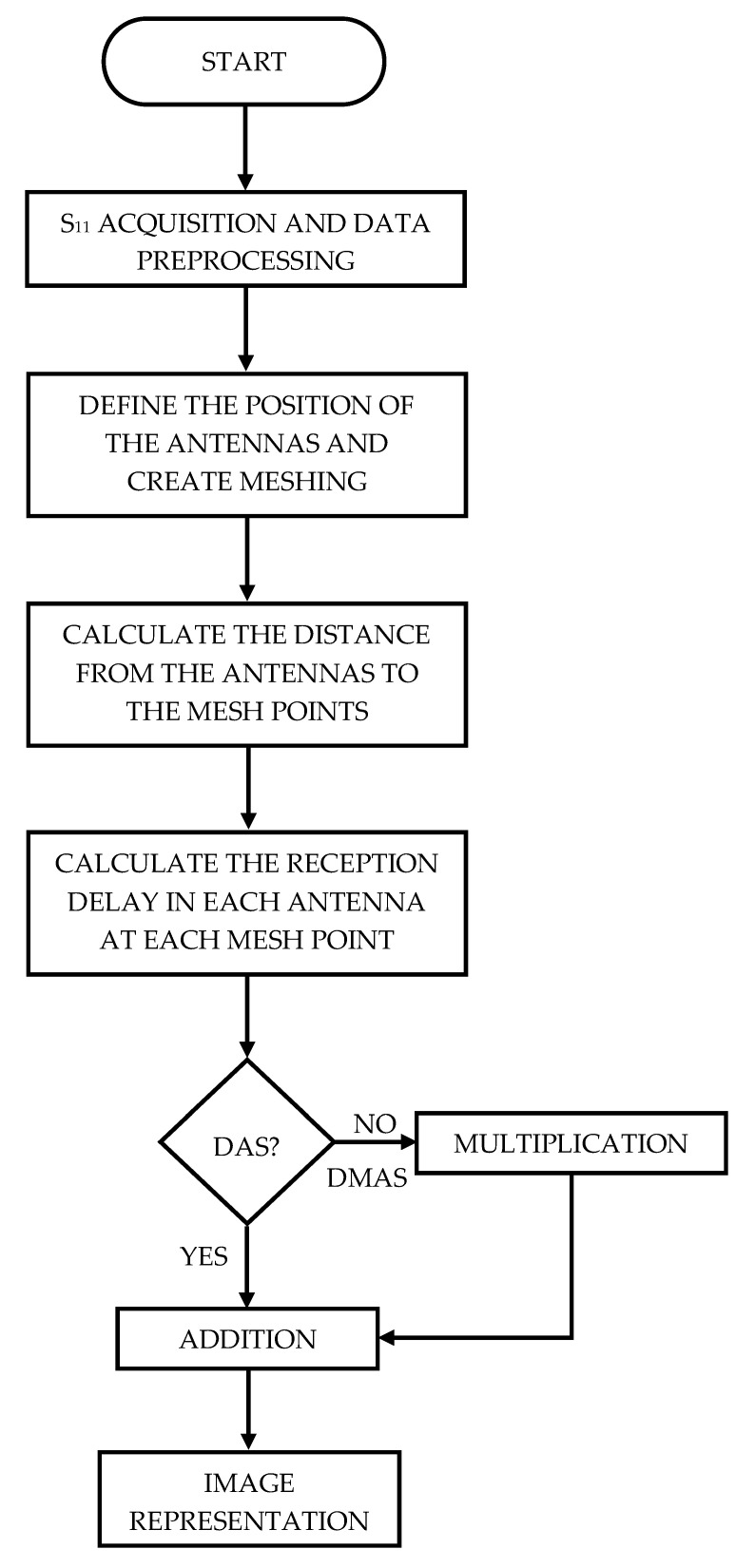
Flowchart for the imaging algorithm applied after each measurement.

**Figure 12 sensors-22-03845-f012:**
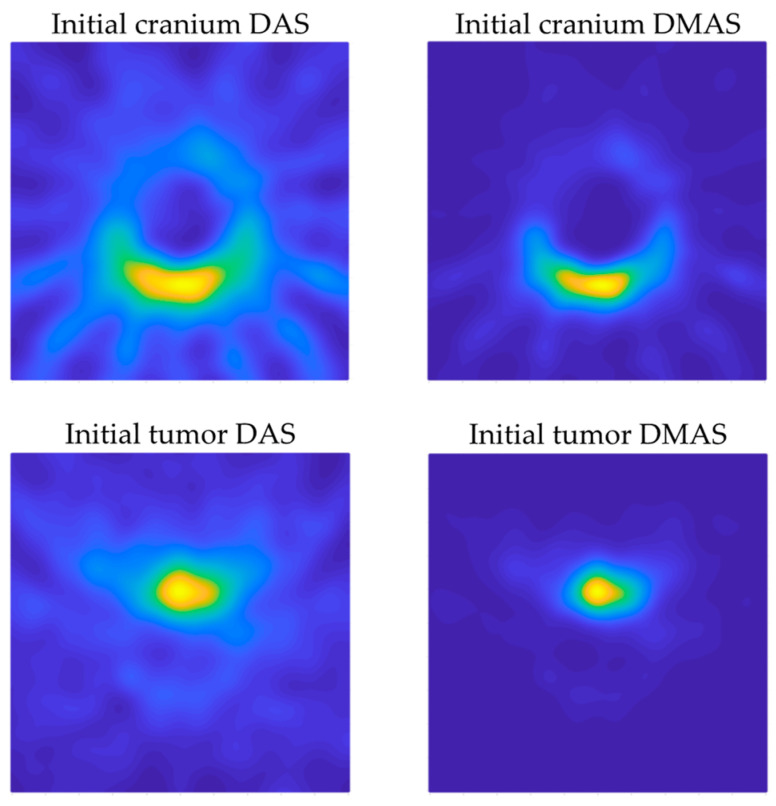
Initial images with DAS and DMAS algorithms.

**Figure 13 sensors-22-03845-f013:**
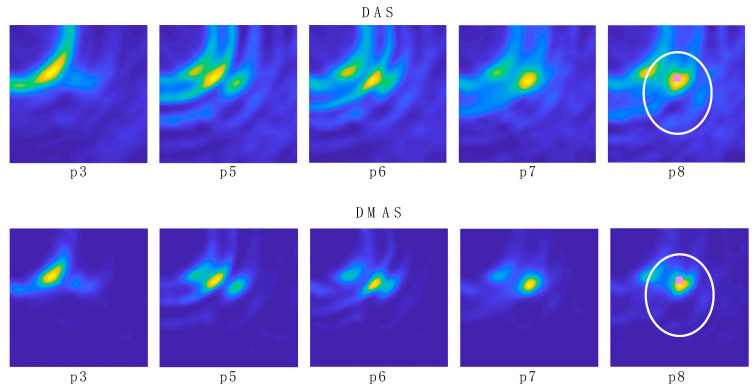
Navigation images for a selection of the positions obtained with DAS and DMAS algorithms. The position and shape of the cranium (white) and tumor (pink)are depicted in the last image for reference.

**Figure 14 sensors-22-03845-f014:**
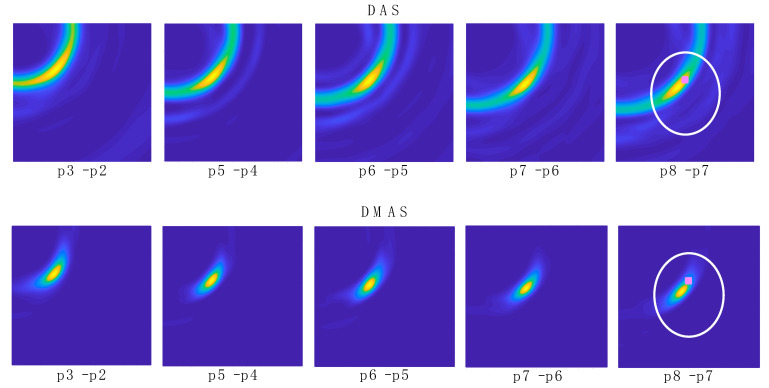
Differential navigation images for a selection of the positions obtained with DAS and DMAS algorithms. The position and shape of the cranium (white) and tumor (pink) are depicted in the last image for reference.

**Figure 15 sensors-22-03845-f015:**
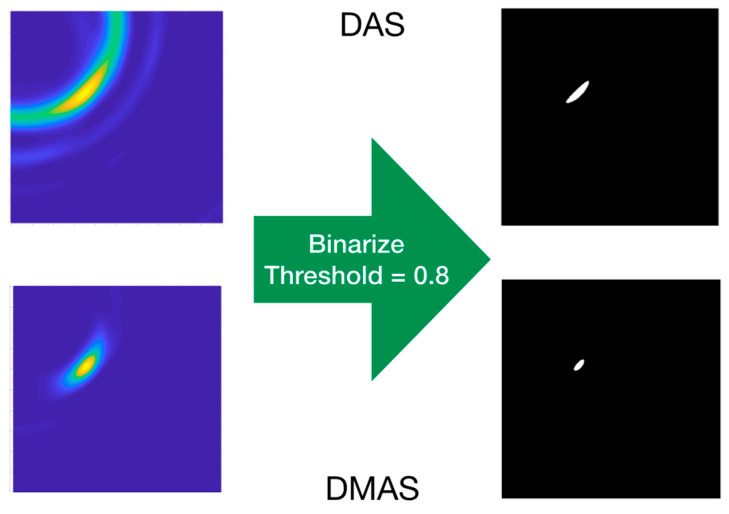
Example of the binarization process for “p5–p4” images.

**Figure 16 sensors-22-03845-f016:**
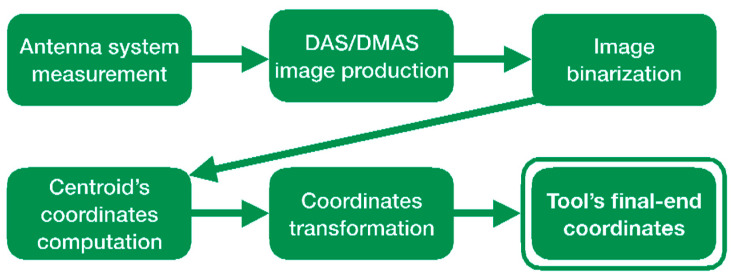
Scheme of the coordinate detection process.

**Figure 17 sensors-22-03845-f017:**
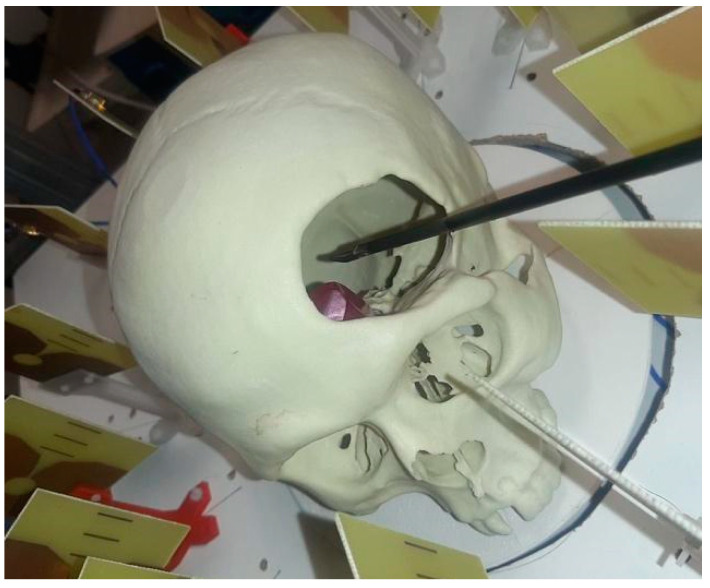
Tool inside the cranium at position p7.

**Figure 18 sensors-22-03845-f018:**
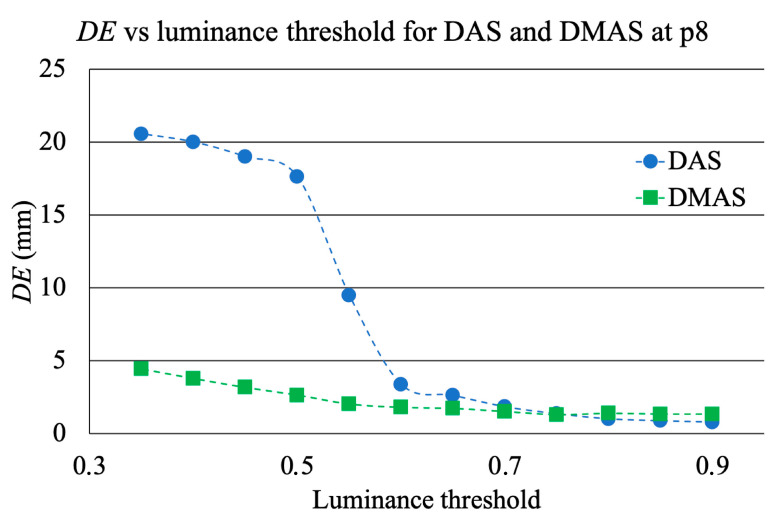
Evolution of *DE* depending on the luminance threshold for both algorithms.

**Table 1 sensors-22-03845-t001:** Detected tool’s final-end coordinates (in mm).

Positions	DAS		DMAS		Robot	
	*x*	*y*	*x*	*y*	*x*	*y*
Tumor	−0.6342	26.1538	0.0961	26.3117	—	—
p2	80.6823	0.4284	111.0871	−13.9562	90.4000	96.2000
p3	47.4907	47.0544	47.4907	44.3263	67.0000	70.8000
p4	42.3399	37.4496	41.8595	36.3899	58.6000	61.7000
p5	36.3243	32.8501	36.1898	31.7679	45.8000	47.9000
p6	26.1574	25.5000	26.1189	24.0119	29.5000	30.2000
p7	17.7778	17.4735	17.8162	16.4589	19.6000	19.5000
p8	11.9544	10.9576	12.4925	10.5743	11.1000	10.4000

**Table 2 sensors-22-03845-t002:** Error analysis for both algorithms (data in mm).

Position Range	DAS				DMAS			
	$\bar{\Delta x}$	$\bar{\Delta y}$	*σ_x_*	*σ_y_*	$\bar{\Delta x}$	$\bar{\Delta y}$	*σ_x_*	*σ_y_*
p3 to p8	8.2593	11.5358	8.2550	11.0136	8.2721	12.8285	8.4759	11.5053
p5 to p8	3.4465	5.3047	4.3779	6.8422	3.3456	6.2968	4.6237	7.0526
p6 to p8	1.4368	2.0563	2.1249	2.6289	1.2575	3.0183	2.4299	3.1812

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
