# Peer review of "Validation of an RF Image System for Real-Time Tracking Neurosurgical Tools"

_sensors, 2022, doi:10.3390/s22103845_

Round 1

Reviewer 1 Report

The paper is very interesting and well presented. However, I have a major concern which authors should try to address.

What happens if you use more realistic phantoms involving human-mimicking tissues? Is the position accuracy affected by the unknown dielectric properties of the scenario? and how? how could you mitigate?

It will be good to add results on this, or at least a detailed discussion.   

Author Response

Please, see attachment

Reviewer 2 Report

The paper proposed a very interesting subject that can be solved by microwave imaging.

However, there are several concerns regarding imaging and results 

1- There are cross talk between antennas that can be presented as Sij such as S12 or erc

however the Authers just provided the S11 and patterns at situation that antenna not used practically

2-Such cross talk are really important because this will be source distortions and blurring of imaging, authors required to study it and propose it in imaging equations.

3- How Authors find out number of antennas used for imaging are optimum and should not be more or less (Adding spatial radiation elements for improvement of imaging resolution)

4-detection error at Figure 17 for soft tissues like human brain seems so high. Please provide more accurate figure and label what’s minimum error and why it’s acceptable.

5- Flow chart or Algorithm of imaging used is required.

Round 2

Reviewer 2 Report

The authors addressed my concerns.